# Associations between Primary Healthcare Experiences and Glycemic Control Status in Patients with Diabetes: Results from the Greater Bay Area Study, China

**DOI:** 10.3390/ijerph20021120

**Published:** 2023-01-08

**Authors:** Junfeng Lu, Hui Yang, Leiyu Shi, Xia Sheng, Yongjun Huo, Ruqing Liu, Ruwei Hu

**Affiliations:** 1Department of Health Management, School of Public Health, Sun Yat-sen University, Guangzhou 510080, China; 2Department of Biostatistics, School of Public Health, and The Key Laboratory of Public Health Safety of Ministry of Education, Fudan University, Shanghai 200032, China; 3John Hopkins School of Public Health, Baltimore, MD 21205, USA; 4Guangdong Provincial Engineering Technology Research Center of Environmental Pollution and Health Risk Assessment, Department of Occupational and Environmental Health, School of Public Health, Sun Yat-sen University, Guangzhou 510080, China

**Keywords:** primary care, chronic diseases, community health centers, patients experiences, diabetes management, community-based intervention

## Abstract

Primary healthcare (PHC) plays an important role in diabetes management; community health centers (CHCs) serve as the main providers of PHC. However, few studies have discussed the association between the service quality of PHC and the effects of diabetes management. In this study, we explored the associations between experiences of PHC in CHCs and glycemic control status in patients with diabetes mellitus. This study was conducted in six CHCs in the Greater Bay Area of China. In total, 418 patients with diabetes mellitus (44% males and 56% females) were recruited between August and October 2019. We evaluated their PHC experiences using the Primary Care Assessment Tool (PCAT) developed by Johns Hopkins and assessed their glycemic control status by measuring their fasting plasma glucose levels. Binary logistic regression analyses were conducted to assess the associations between the patients’ PHC experiences and glycemic control status, adjusting for covariates. The patients with good glycemic control had significantly higher total and dimensional PCAT scores compared with those with poor glycemic control (*p* < 0.05). Higher PCAT scores were significantly associated with a greater adjusted odds ratios (aORs) of good glycemic control for total and dimensional PCAT scores. For example, compared to those with poor glycemic control, the aORs for those with good glycemic control was 8.82 (95% CI = 4.38–17.76) per total PCAT score increasing. Especially, the aORs for those with good glycemic control were 3.92 (95% CI = 2.38–6.44) and 4.73 (95% CI = 2.73–8.20) per dimensional PCAT score of family-centeredness and community orientation increasing, respectively. Better PHC experiences were associated with better diabetes management. In particular, family-centered and community-oriented CHCs may help improve diabetes management in China and other low- and middle-income countries.

## 1. Introduction

In 2021, the World Health Organization (WHO) raised the priority given to the prevention, control, and management of diabetes as well as its risk factors with the aim of expediting the necessary actions for diabetes management moving forward, 100 years after the discovery of insulin [1,2]. According to the report released by the International Diabetes Federation [3] in 2019, ~463 million adults aged between 20 and 79 years worldwide, constituting ~6% of the global population, are struggling with type 1 or 2 diabetes mellitus (DM); this indicates that 1 in 11 people has diabetes globally. In addition, early deaths due to diabetes were noted to increase by 5% between 2000 and 2016, whereas premature mortality due to other major noncommunicable diseases is decreasing [2]. Even worse is that people living with DM are at increased risk of severe illness and death due to COVID-19. Undoubtedly, DM has emerged as a great challenge to all countries, particularly low- and middle-income countries (LMICs) [4]. As the prevalence rate of diabetes is expected to increase significantly in middle-income countries over the next 25 years [5], health policy-makers in LMICs should find appropriate solutions to manage the disease.

As the largest LMIC in the world, China has faced the formidable challenge of an increasing burden of DM, along with changes in lifestyle, an aging population, and environmental issues due to rapid economic development and urbanization over the past 40 years. During rapid economic development and urbanization, people are, on average, living in more precarious conditions, less educated, and less attentive to potential health problems. Therefore, they tend to have higher than average rates of overweight, obesity, and alcohol consumption [6]. In fact, the country has accounted for the highest number of people with diabetes (114 million) worldwide in 2019 [3]. Medical management of diabetes alone, without complications, is already estimated to represent 8.5% of the national health expenditure in China [6]. These data collectively indicate that the solution to the crisis of diabetes in China may galvanize efforts around the world to reduce the risk of diabetes.

Studies have shown that primary healthcare (PHC) can enable patients to access more appropriate medical management, improve patients’ quality of life and satisfaction, promote healthcare equity, and reduce health expenditure and the utilization of hospitalization and specialist services [7,8,9,10,11,12,13,14,15,16]. PHC refers to first contact, continuous, comprehensive, and coordinated care provided to individuals regardless of sex, disease, or organ system affected [17]. Diabetes has been considered an ambulatory care sensitive condition, for which the risk of hospitalization can be reduced through timely and effective PHC services [7]. These studies suggest that diabetes may be addressed from the viewpoint of improving DM patients’ experiences of PHC, not just from the clinical viewpoint.

In fact, in China’s latest round of health system reform since 2009, enormous efforts have been made to revitalize the country’s PHC program, with the vision of forging a PHC-based integrated delivery system where PHC facilities serve as gatekeepers and secondary and tertiary hospitals provide specialized care [8,9]. For better implementation of the reform, the Chinese government has put forward a series of policies and pilot programs. First, it launched the Basic Public Health Service Program, which lists diabetes as a key condition for chronic disease management [10]. Second, it established a universal health insurance coverage program and the National Essential Medicine Program, which reduce the financial burden and provide fundamental medical security for patients [11]. Third, it implemented the Family Doctor Program, which provides effective health management to those who have signed up with a family doctor [12]. Lastly, it enacted a series of policies to implement the core functions of PHC in community health centers (CHCs) since the first official national policy in 1999 and especially improved the quality and quantity of CHCs since the health reform of 2009 [13]. These initiatives have clearly shown that the government has placed great importance on PHC in CHCs since 2009; however, the outcomes of the policies are not as good as expected, with many problems remaining unresolved. In addition, the number of people with diabetes in China has been increasing since 2009, and it is anticipated to continue to rise until 2030 [14], indicating that the health policy on PHC in China still does not match the needs of addressing diabetes. In line with the statement of Sun et al. [7], “where the quality of care is low, service delivery is weak, and access is limited, patients are likely to bypass primary care services,” the diabetes management policies in China should try to improve DM patients’ PHC experiences, such as by improving their experiences in CHCs.

The emerging studies have found that patients with a chronic disease have better experiences in PHC facilities [15,16,18], but few studies assess the relationship between DM patients’ PHC experiences and their glycemic control status. Evaluating patients’ experiences from the perspective of PHC should be able to provide more comprehensive and targeted line of evidence to policy-makers. Meanwhile, CHCs are the main providers of diabetes management in China. Therefore, in this study, we aimed to evaluate various aspects of DM patients’ PHC experiences in CHCs, such as first contact, continuous, comprehensive, and coordinated care, and to explore the relationship between patients’ PHC experiences and glycemic control status. The findings of this study have implications on the prevention and control of diabetes, and on the prospective PHC reforms in China.

## 2. Materials and Methods

The study population was derived from the Greater Bay Area Study [19,20]. A cross-sectional investigation carried out from August to October 2019 in selected urban CHCs in Guangzhou, the largest metropolis in southern China. The design of the Greater Bay Area Study and its eligibility criteria have been previously described in detail as shown in Figure 1 [19]. In brief, we employed a multistage stratified clustering sampling protocol. Six CHCs were randomly selected from four center-urban districts, namely, Liwan, Yuexiu, Tianhe, and Haizhu. One family physician group was then randomly selected from each selected CHC. In the final sampling stage, participants were randomly recruited by the selected family physician group during their visits to the CHC. Thereafter, participants with diabetes who were aged over 20 years old without auditory or visual impairment, mental illness, or other issues that might interfere with study participation and had been a resident of the district for at least 1 year were recruited. DM patients with severe mental health disorders or those who could not understand the questionnaires were also excluded. Of the 418 patients, 336 returned valid questionnaires, giving an effective response rate of 80.38%.

Fasting plasma glucose (FPG) levels were measured to assess the glycemic control status of the patients. The patients were divided into two groups, namely, the good glycemic control group (4.4 mmol/L ≤ FPG ≤ 7.0 mmol/L) and the poor glycemic control group (FPG < 4.4 mmol/L or FPG > 7.0 mmol/L). This stratification was based on the *Guidelines for the Prevention and Control of Type-2 Diabetes in China* (*2017 Edition*) [21].

A Chinese version of the Primary Care Assessment Tool (PCAT)—Short Edition (Johns Hopkins, Washington D.C., United States), a tool developed by the Johns Hopkins Primary Care Center under the leadership of Professors Barbara Starfield and Leiyu Shi [19], was created and validated to measure the patients’ PHC experiences. This scale included the following seven main dimensions: (1) first contact, refers to access to and use of primary care services when a new health or medical problem arises; (2) continuity of care, the longitudinal use of a regular source of primary care over time; (3) coordination, the interpersonal linkage of care between different levels of providers or informational linkage of care through electronic information systems; (4) comprehensiveness, the availability of clinical and preventive services from the provider or provider group; (5) family-centeredness, inclusion of family health concerns in decision-making; (6) community orientation, the provider’s knowledge of community health needs; and (7) cultural competence, patients’ willingness to recommend their primary care provider to others. Furthermore, these seven dimensions can be divided into 11 sub-dimensions: A. Extent of affiliation with a place/doctor; B. first contact—utilization (the extent to which the primary care provider performs a gatekeeper function); C. first contact—access (whether patients can contact a physician in time when they need medical and health services); D. ongoing care (the continuous relationship between physicians and patients in primary care institutions); E. coordination—care (the interpersonal linkage of care among different levels of providers); F. coordination—information systems (informational linkage of care through the use of an electronic information system); G. comprehensiveness—services available (the ability to perform a wide range of services in primary care); H. comprehensiveness—services used (the appropriate provision of services during consultations by a primary care provider); I. family-centeredness (the recognition of the family as a major participant in the diagnosis, treatment, and recovery of patients); J. community orientation (whether CHCs fully consider the needs of patients in the implementation of health services); and K. cultural competence (the provision of care that respects the beliefs, interpersonal styles, attitudes and behaviors of people as they influence health) [19]. These 11 dimensions consist of 35 items, with each item scored on a 4-point Likert scale (1 = never; 2 = sometimes; 3 = often; 4 = always). A “do not know/not sure” response was assigned a neutral value of 2.5 [22]. The score for each dimension was the average of the values for all the items under that dimension. The total PCAT score was the sum of the values for all dimensions. The higher the score, the better the experiences were.

Sociodemographic and health management data were collected using a self-reported questionnaire. Sociodemographic covariates included sex, age, education, annual family income, and residence. Health management information included health insurance, health status, and satisfaction with CHCs. Age was classified as ≤60 years old or >60 years old. Education level was categorized as primary school or lower, middle school, high school, or undergraduate or higher. Annual family income was grouped into ≤80,000, 80,000–150,000, 150,000–200,000, and >200,000 yuan according to quartile of household income. Residence was classified as local or nonlocal. Insurance was classified as employee or resident. Health status was classified as poor, moderate, or good. Satisfaction with CHCs was grouped into not satisfied or satisfied.

The quantitative variables were expressed as means employee median (IQR), and the categorical variables were expressed as absolute numbers (n) and percentages (%). We used Pearson’s chi-square test to compare demographic characteristics between poor and good glycemic control groups and used standardized z-score to compare PCAT scores between poor and good glycemic control groups.

Binary logistic regression analyses were conducted to assess the relationship between the patients’ PHC experiences and glycemic control status. Age, sex, education, household income, residence, health status, and satisfaction with CHCs were introduced as covariates in the adjusted models [18,19,21,22,23,24,25]. The dependent variables used in the models were PCAT score, first contact—utilization score, first contact—access score, ongoing care score, coordination—care score, coordination—information systems score, comprehensiveness—services available score, comprehensiveness—services used score, family-centeredness score, community orientation score, and cultural competence score. The glycemic control status effect was reported using adjusted beta with 95% CIs. A two-sided *p*-value < 0.05 was considered significant. All statistical analyses were performed using IBM SPSS 22.0.

We obtained written informed consent from each participant before data and sample collection. The study procedure was approved by the Human Studies Committee of Sun Yat-sen university in compliance with the Declaration of Helsinki (No.IRB2014.9).

## 3. Results

The characteristics of the study population are summarized in Table 1. Most of the patients were > 60 years old (83.63%). Of all patients, with 44.00% males and 56.00% females, 92.56% were local residents, 83.63% had employee health insurance, 26.79% had poor glycemic control, and 73.21% had good glycemic control. Compared to the diabetes patients with poor glycemic control, more local residents were found among those with good glycemic control (85.56% vs. 95.12%, *p* < 0.05). Significantly more patients with good glycemic control have employee health insurance than those with poor glycemic control (88.62% vs. 70.00%, *p* < 0.05). More patients with good glycemic control were satisfied with CHCs than those with poor glycemic control (94.72% vs. 87.78%, *p* < 0.05).

Table 2 shows the median (IQR) of PCAT scores in the good glycemic control and poor glycemic control groups. The total PCAT score of the poor glycemic control group was significantly lower than that of the good control group (2.72 (0.37) vs. 3.78 (0.96), respectively, *p* <0.001) as were all dimensional scores. Notably, the scores of the first contact—utilization dimension were the highest in both groups, while the scores of the first contact—access dimension were the lowest in both groups.

Table 3 demonstrates the association between glycemic control status and PCAT scores. In the adjusted logistic model, higher PCAT scores were significantly associated with greater adjusted odds ratios (aORs) of good glycemic control for total and dimensional PCAT scores. For example, the aOR for having good glycemic control was 8.82 (95% CI = 4.38–17.76) per total PCAT score. The aORs for having good glycemic control were 3.92 (95% CI = 2.38–6.44) and 4.73 (95% CI = 2.73–8.20) per dimensional PCAT score of family-centeredness and community orientation, respectively.

## 4. Discussion

In this study, we found positive associations between experiences of PHC in CHCs and glycemic control status in patients with diabetes. Our results provided additional evidence that diabetes may be addressed more effectively from improving DM patients’ experiences of PHC. To the best of our knowledge, this study is the first one published to analyze the association between DM patients’ experiences in CHCs and glycemic control status in China.

Consistent with the findings of previous studies [19,20,23,26,27,28,29,30,31], in this study, CHCs performed well in providing PHC to DM patients, especially in first contact care utilization. This may be partly attributable to the significance placed by the Chinese government on CHCs from 2009 to 2016, with nearly half of the health policies enacted by the government being targeted at CHCs [13]. CHCs in China are now mostly financed by government subsidies to deliver PHC [32] and enjoyed specific or targeted policies such as through the National Essential Public Health Service Program. The CHC model is appropriate for the delivery of PHC [23] by providing accessible, cost-effective, and high-quality PHC and reducing health disparities [33,34,35,36].

The first contact—utilization dimension of the PCAT has been used to evaluate whether patients will prioritize choosing CHCs for their PHC. The higher PCAT scores for this dimension in this study are related to the implementation of the gatekeeping policy in China [27]. This policy requires patients to make CHCs their first contact of care; without a referral from designated CHCs, the rate of health insurance reimbursement for tertiary hospitals would be reduced, which would, in turn, contribute to reducing the utilization of hospitalization and specialist services [37,38] as well as the hospitalization rate among patients with diabetes in China [7]. Based on this premise, this policy was helpful in guiding DM patients to seek CHC services first.

The first contact—access dimension of the PCAT is used to evaluate whether patients are able to access PHC services in CHCs in a timely manner. Compared with some developed countries where DM patients need to schedule an appointment first [39], in China, such patients can immediately reach out to CHCs for treatment. The lower scores for items under the first contact—access dimension in this study—such as “When your Primary Care Physician (PCP) is closed, is there a phone number you can call when you get sick?” and “When your PCP is open, can you get advice quickly over the phone if you need it?”—indicate that DM patients could only access CHCs for onsite treatment, instead of accessing PHC services through mobile management.

Evidently, DM patients in China are likely to choose CHCs for their first contact of care, but the accessibility of these facilities remains limited. The Family Doctor Program was shown to be valid in improving the quality of PHC at first contact [40] and for chronic condition management [41]. However, the free service package provided by family doctor teams [40] needs to be enhanced; in addition, mobile management, as well as after-hours diagnostic and treatment services, need to be reinforced. We recommend that family doctor teams use websites, mobile apps, and other media to establish a platform, such as a WeChat or Telegram group, to co-manage DM patients. Each team would be responsible for a specific period to form a 24-hour network that provides PHC services. Such a system would allow DM patients to seek diabetes management advice as well as to secure after-hours diagnostic and treatment services through the said platform. In addition, we suggest that family doctor groups reach out to DM patients through their family units. This system would then be particularly beneficial when family doctors need to monitor DM patients’ health conditions and remind them to adhere to their diabetes management protocol via a mobile application, since the patient, for instance, an older adult, is not familiar or comfortable with mobile technology. It would also allow DM patients to contact their family doctor first when they need PHC services to obtain more precise DM management services.

In addition to first contact related dimensions, another two dimensions merit special attention, the family-centeredness and community orientation dimensions. The former dimension is used to evaluate the relationship of PHC providers with the family of their patient, whereas the latter one is used to evaluate the relationship between CHCs and residents in the community.

In the present study, the scores for the dimensions of family-centeredness and community orientation were both lower than those for the other dimensions, which is consistent with the findings of previous studies [26,42,43,44]. On one hand, the lower scores for the item “Does your PCP ask you about your ideas and opinions when planning treatment and care for you or a family member?” under the family-centeredness dimension suggest that CHC service providers would not seek advice from patients or their family members when making treatment plans owing to communication barriers or insufficient medical knowledge, although many experts call for greater autonomy among and participation from patients [8,45]. As a result, it is difficult for patients to achieve equality in their diabetes management decisions. On the other hand, the lower scores for the item “Does your PCP get opinions and ideas from people that will help to provide better health care?” under the community orientation dimension can be attributed to the fact that China’s current CHCs are still treatment-centered, rather than population- and community-based [23,46,47], which leads to the challenge of transitioning from an approach based on diagnosis and treatment to one based on prevention and control [23].

In contrast, with the higher aORs, the lower PCAT scores for both family-centeredness and community orientation indicate that CHCs should be not only population-based [48] but also family-centered and community-oriented, to slow down the deterioration of FPG control caused by social distance and isolation during the COVID-19 pandemic or other reasons. By providing care in and through the community, family- and community-oriented CHCs would be able to address not only individual and family health needs but also the broader issue of public health and the needs of defined populations so as to improve patient access to PHC services, reduce hospitalizations, as well as enhance the cost-effectiveness and equity of healthcare [49].

First, each DM patient has a unique profile of risk factors and complications, which is why CHCs should be population-based [48] in addressing the broader determinants of DM health needs through multi-sectoral policy and action, thereby empowering DM patients in CHCs to take charge of their own health and allowing for more targeted diabetes treatment. Second, under the Family Doctor Program, diabetes management or intervention programs and health education should be family-oriented when formulated by family doctors from CHCs [7]. Such measures in family units are supervised by family members; they are beneficial in involving family members’ participation in addressing patient’s health behaviors and risk factors of diabetes, which could ultimately help improve the patient’s health condition and reduce the incidence of diabetes. Third, the community-based model requires CHCs to periodically launch health education or other campaigns in the community to increase patients’ social recognition and their sense of belongingness as well as to shift their demand for PHC toward CHCs. Basically, CHCs can recruit volunteers from the community through social recognition incentives to reduce the burden on health campaigns and the daily operation of CHCs.

In line with the above-described challenges of and achievements in diabetes management, we provide several policy recommendations for China and other LMICs with similar circumstances: First, the entry point for diabetes management policies should be at the PHC level, which provides both preventive and control services within communities and close to DM patients throughout their lifespan, thus ensuring effective and inexpensive diabetes management. Second, the government should encourage, empower, and finance CHCs in providing a higher quality of PHC services [7]. The government should also increase the fixed salary of CHC workers and provide CHCs with financial incentives based on region and performance indexes, such as health campaigns, patients’ satisfaction, and policy completion. Third, health insurance policies should give higher financing priority to CHCs—for example, by removing such barriers as low reimbursement caps on PHC services [8]. Finally yet importantly, policy-makers should promote family-centered and community-oriented CHCs. This would guarantee healthcare equity for all individuals, not only DM patients.

This study used the PCAT innovatively to evaluate the DM patients’ experiences in CHCs services and to provide valuable evidence that the role of CHCs can enhance the health condition of DM patients and address diabetes at low health expenditure levels. In addition, the findings of this study highlight the need for policy-makers not only in China but also in other LMICs to promote the implementation and dissemination of PHC policies. However, several study limitations should be considered. First, we used only FPG analysis to assess glycemic control status, which may not comprehensively reflect DM patients’ multifaceted health conditions. Second, because of the cross-sectional design of this study, we were not able to establish cause–effect relationships and recall bias could have existed. Third, the participants with auditory or visual impairment, mental illness, or other issues that might interfere with study participation were excluded from our research. This group is probably more likely to have poorer, or at the very least different, primary care experiences, so an analysis incorporating the persons with disabilities might yield different results. Fourth, the study lacked representativeness since the study was carried out in one metropolitan city in China. Further studies should be carried out in more cities and regions across China.

## 5. Conclusions

The results of this study provide a perspective on and supporting evidence for diabetes policies and programs that focus on the prevention and control of diabetes, instead of the diagnosis and treatment thereof, in CHCs. On one hand, China has made substantial progress in PHC reform, and DM patients in China tend to choose CHCs for their first contact of care. However, mobile channels through which they could access PHC remain blocked, which the next reform should factor in. On the other hand, we found that DM patients’ glycemic control status was associated with their PHC experiences, although their experiences of family-centeredness and community orientation warrant further improvement. Thus, family-centered and community-oriented CHCs are recommended for China and other LMICs that share the challenge of improving the quality of PHC and addressing diabetes.

## Figures and Tables

**Figure 1 ijerph-20-01120-f001:**
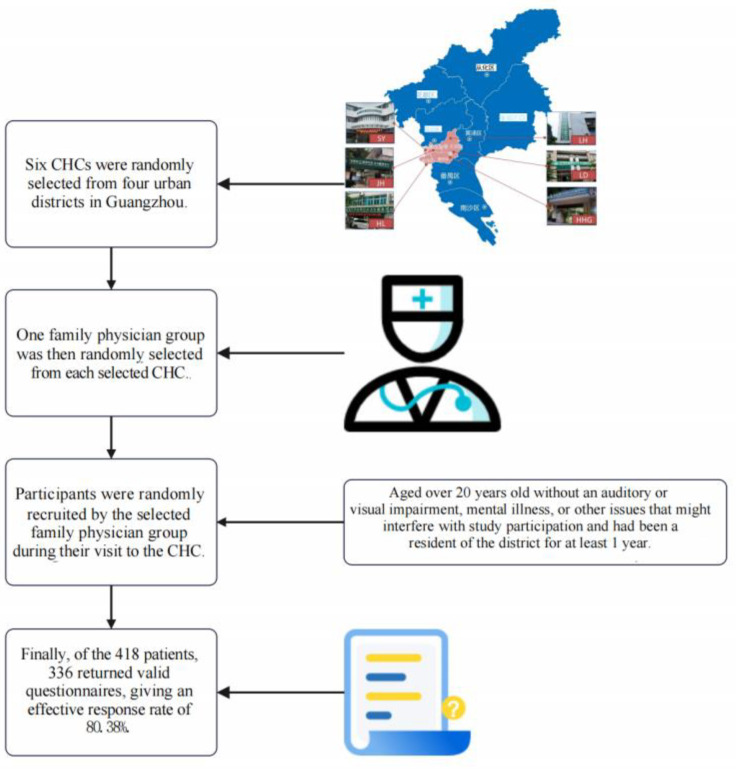
Study flow diagram.

**Table 1 ijerph-20-01120-t001:** Baseline characteristics of the study population (n = 336).

Variable	Total Population (n = 336)	Good Glycemic Control Group (n = 246)	Poor Glycemic Control Group (n = 90)	*p*
Sex				
Male	148 (44.00)	108 (43.90)	40 (44.44)	0.929
Female	188 (56.00)	138 (56.10)	50 (55.56)	
Age, y				
≤60	55 (16.37)	45 (18.29)	10 (11.11)	0.115
>60	281 (83.63)	201 (81.71)	80 (88.89)	
Education				
Primary school or lower	54 (16.07)	39 (15.85)	15 (16.67)	0.301
Middle school	111 (33.04)	75 (30.49)	36 (40.00)	
High school	130 (38.69)	102 (41.46)	28 (31.11)	
Undergraduate or higher	41 (12.20)	30 (12.20)	11 (12.22)	
Annual family income, yuan				
≤80,000	75 (22.32)	59 (23.98)	16 (17.78)	0.031
80,000–150,000	88 (26.19)	62 (25.20)	26 (28.89)	
150,000–200,000	66 (19.64)	40 (16.26)	26 (28.89)	
>200,000	107 (31.85)	85 (34.55)	22 (24.44)	
Nonlocal	25 (7.44)	12 (4.88)	13 (14.44)	
Insurance				
Employee	281 (83.63)	218 (88.62)	63 (70.00)	<0.001
Resident	55 (16.37)	28 (11.38)	27 (30.00)	
Health status				
Poor	34 (10.12)	17 (6.91)	17 (18.89)	<0.001
Moderate	234 (69.64)	186 (75.61)	48 (53.33)	
Good	68 (20.24)	43 (17.48)	25 (27.78)	
Satisfaction with CHCs				
Not satisfied	24 (7.14)	13 (5.28)	11 (12.22)	0.029
Satisfied	312 (92.86)	233 (94.72)	79 (87.78)	

Abbreviations: CHCs, community health centers. Data are presented as n (%). *p*-values are based on 2 tests.

**Table 2 ijerph-20-01120-t002:** PCAT scores stratified by glycemic control status.

Dimension	Total Population (n = 336)	Poor Glycemic Control Group (n = 90)	Good Glycemic Control Group (n = 246)	*Z*	*p*
First contact—utilization	3.44 (0.63)	3.07 (0.64)	3.58 (0.57)	−6.52	<0.001
First contact—access	3.01 (0.74)	2.45 (0.60)	3.22 (0.68)	−8.33	<0.001
Ongoing care	3.25 (0.76)	2.63 (0.65)	3.48 (0.66)	−8.87	<0.001
Coordination—care	3.32 (0.69)	2.83 (0.59)	3.50 (0.64)	−8.31	<0.001
Coordination—information systems	3.37 (0.78)	2.76 (0.71)	3.59 (0.68)	−9.35	<0.001
Comprehensiveness—services available	3.43 (0.66)	2.96 (0.67)	3.60 (0.57)	−8.01	<0.001
Comprehensiveness—services used	3.32 (0.67)	2.86 (0.51)	3.49 (0.64)	−7.86	<0.001
Family-centeredness	3.29 (0.74)	2.73 (0.63)	3.5 (0.66)	−8.52	<0.001
Community orientation	3.15 (0.71)	2.63 (0.52)	3.34 (0.67)	−8.5	<0.001
Cultural competence	3.31 (0.78)	2.81 (0.68)	3.50 (0.73)	−7.78	<0.001
Total PCAT score	3.29 (0.61)	2.78 (4.18)	3.48 (5.63)	−8.85	< 0.001

Abbreviation: PCAT, Primary Care Assessment Tool. Data are presented as median (IQR). Good glycemic control was defined as 4.40 mmol/L ≤ fasting plasma glucose ≤ 7.00 mmol/L, whereas poor glycemic control was defined as fasting plasma glucose < 4.40 mmol/L or fasting plasma glucose > 7.00 mmol/L. *p*-values are based on non-parametric test.

**Table 3 ijerph-20-01120-t003:** PCAT scores associated with glycemic control status.

Dimension	COR (95% CI)	*p*	AOR (95% CI) ^a^	*p*
First contact—utilization	3.48 (2.32–5.21)	< 0.001	3.33 (2.18–5.08)	<0.001
First contact—access	5.47 (3.50–8.54)	< 0.001	5.09 (3.23–8.01)	<0.001
Ongoing Care	5.48 (3.58–8.40)	< 0.001	5.58 (3.55–8.77)	<0.001
Coordination—care	4.55 (3.00–6.91)	< 0.001	4.38 (2.85–6.74)	<0.001
Coordination—information systems	4.16 (2.89–5.99)	< 0.001	3.91 (2.68–5.70)	<0.001
Comprehensiveness—services available	4.38 (2.90–6.60)	< 0.001	4.25 (2.80–6.47)	<0.001
Comprehensiveness—services used	4.44 (2.93–6.73)	< 0.001	4.04 (2.64–6.18)	<0.001
Family-centeredness	4.76 (3.17–7.13)	< 0.001	4.46 (2.95–6.76)	<0.001
Community orientation	5.12 (3.28–8.00)	< 0.001	4.90 (3.11–7.71)	<0.001
Cultural competence	3.14 (2.23–4.42)	< 0.001	2.98 (2.09–4.24)	<0.001
Total PCAT score	9.55 (5.53–16.50)	< 0.001	9.77 (5.59–17.08)	<0.001

Abbreviations: AOR, adjusted (age, sex, health status, and satisfaction with community health centers) odds ratio; COR, crude odds ratio; PCAT, Primary Care Assessment Tool. ^a^ Adjusted for age, sex, education, household income, residence, health status, insurance, and satisfaction with community health centers in the logistic models.

## Data Availability

The data presented in this study are available on request from the corresponding author. The data are not publicly available due to ethical and privacy concerns.

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
