# Peer review of "Associations between Primary Healthcare Experiences and Glycemic Control Status in Patients with Diabetes: Results from the Greater Bay Area Study, China"

_ijerph, 2023, doi:10.3390/ijerph20021120_

Round 1

Reviewer 1 Report

Overall comments

I congratulate the authors on an important paper on the relevance of primary health care to diabetes management. The study appears to be conducted with sound methodology and the conclusions generally match the findings. However, I would lie to caution the authors against over generalization of the applicability of the findings to have implications for all of China and even the rest of the world based on what is a small study in a select geographical location. This is implied in authors’ statements in discussion and introduction.

Below I have outlined more specific recommended changes that I hope would improve the quality of what is already a good article.

Introduction

Line 66-67- “These data collectively indicate that the solution to the crisis of diabetes in China 66

may significantly help other countries manage diabetes.”

This statement makes a leap not supported by previous statements. Solutions to diabetes crisis in China may help other countries or it may not. The sentence should reflect the uncertainty in the statement by indicating it is a hypothesis. The quoting of statistics from China is not an indicator that data from China will help other very different LMICs.

Line 93: Sentence “however, the outcomes of the policies are not as good as willed, with many problems remain unresolved.” 

To improve fluency this can be replaced with “however, the outcomes of the policies are not as good as expected, with many problems remaining unresolved.”

Methods

Lines 121-123: Thereafter, participants with diabetes who were aged over 20 years old without an auditory or visual impairment, mental illness, or other issues that might interfere with study participation and had been a resident of the district for at least 1 year were recruited.

The exclusion of persons with disabilities is an important one given that this group is probably more likely to have poorer at the very least different primary care experiences. I recommend that the authors mention the limitation of generalizing to this population in their discussion and also discuss how their exclusion may have affected the findings.  

Line 176: Kindly indicate which non-parametric tests were used and for which outcome variables. Similarly, which for which variables did you use Chi-squared?

Line 180: State the rationale for including these particular variables in the model- a priori hypothesis or based on hierarchical statistical modelling?

Line 191: The word “obviously” is unnecessary.

Line 199: “So did all dimensional scores”. Advise replacing “did” with “were”

Discussion

Line 238: Authors state: “The first contact–utilization dimension of the PCAT has been used to evaluate whether patients will prioritize choosing CHCs for their PHC.”

It seems to me that if CHCs are mandatory for first contact as the authors state in the next sentence, then this is not a true measure of patients’ priorities. It’s probably more appropriate to consider it a measure of adherence/compliance to the policy.

Reviewer 2 Report

This is an interesting article showing positive associations between experiences of Primary healthcare (PHC) in CHCs and glycemic control status in patients with diabetes, indicating the important role of diabetes policies and programs that focus on the prevention and control of diabetes. The strength of this study is defined by the methods and tool used (eg. a Chinese version of the Primary Care Assessment Tool (PCAT)–Short Edition).  However, I have a few concerns.

1. It requires more detailed information of the background of the study population to make a conclusion, for example, eating habit, exercise habit, and family health history.

2. The study was carried out  in Guangzhou, the largest metropolis in southern China. Is the conclusion applicable in other part of China, especially in the remote area. The author should discuss about it in the paper.

Reviewer 3 Report

The manuscript reviewed entitled "Associations Between Primary Healthcare Experiences and Glycemic Control Status in Patients with Diabetes: Results from the Greater Bay Area Study, China" was written well and have scientific sound. But recommend major revision before the publication.

1. Please fix the grammatical error of this sentence "Primary healthcare (PHC) plays an important role in diabetes management, of which com- 22 munity health centers (CHCs) serve as the main providers".

2. As authors have selected the " six CHCs" can you please explain it why these particular centers selected please give description in the text.

3. Please provide abbreviations of all the non scientific words in a separate section like "(LMICs, LMIC, IQR, PHC).

4. Please reduce the size of material and methods by adding the flow sheet so that the methods becomes clear and short as in present form it is not giving proper sense.

5. As authors are studying the human based study trails please provide the ethical statement of any approved meeting or any international guidelines.

6. I recommend a English correction by a professional English service.

Round 2

Reviewer 3 Report

Authors have done the changes according to suggestions now it can be accepted.